# Synthesis and Characterization of an Analgesic Potential Conotoxin Lv32.1

**DOI:** 10.3390/molecules27238617

**Published:** 2022-12-06

**Authors:** Siyi Liu, Cheng Li, Shen You, Qinghui Yan, Sulan Luo, Ying Fu

**Affiliations:** 1Key Laboratory of Tropical Biological Resources of Ministry of Education, School of Pharmaceutical Sciences, Hainan University, Haikou 570228, China; 2Medical School, Guangxi University, Nanning 530004, China

**Keywords:** analgesic conotoxin, Lv32.1, *Conus lividus*, α9α10 nAChR, Na_v_1.8 channel

## Abstract

In our work of screening analgesic peptides from the conotoxin libraries of diverse *Conus* species, we decoded a peptide sequence from *Conus lividus* and named it Lv32.1 (LvXXXIIA). The folding conditions of linear Lv32.1 on buffer, oxidizing agent, concentration of GSH/GSSG and reaction time were optimized for a maximum yield of (34.94 ± 0.96)%, providing an efficient solution for the synthesis of Lv32.1. Its disulfide connectivity was identified to be 1–3, 2–6, 4–5, which was first reported for the conotoxins with cysteine framework XXXII and different from the common connectivities established for conotoxins with six cysteines. The analgesic effect of Lv32.1 was determined by a hot plate test in mice. An evident increase in the pain threshold with time illustrated that Lv32.1 exhibited analgesic potency. The effects on Na_v_1.8 channel and α9α10 nAChR were detected, but weak inhibition was observed. In this work, we highlight the efficient synthesis, novel disulfide linkage and analgesic potential of Lv32.1, which laid a positive foundation for further development of conotoxin Lv32.1 as an analgesic candidate.

## 1. Introduction

Chronic pain is a suffering, costly and long-term problem that significantly affects society and individuals [1,2]. The International Association for the Study of Pain (IASP) defines it as pain that lasts for longer than 3 months [3]. Chronic pain does not only happen to old people and its morbidity has increased [4] (Larsson et al., 2017). A total of 20.6% of the adolescents across 42 countries have self-reported chronic pain in at least two sites of headache, stomach and backache [5]. Approximately 20% of the adult population have chronic pain [6]. The financial cost for chronic pain was estimated at over USD 200 billion per year in Europe and USD 150 billion per year in the USA [6,7]. Analgesic medication for pain includes the following: (1) nonopioid analgesics such as paracetamol, (2) nonsteroidal anti-inflammatory drugs (NSAIDs) such as aspirin, ibuprofen, celecoxib and etoricoxib, (3) opioid analgesics such as codeine, fentanyl, buprenorphine and naproxen, (4) antidepressants such as tricyclic antidepressants (TCAs), serotonin and noradrenaine reuptake inhibitors (SNRIs), (5) antiepileptic drugs such as pregabalin and carbamazepine, (6) anticonvulsants, such as gabapentin, etc. [8,9]. However, patients rarely achieve long-term pain relief due to side effects such as cardiovascular, gastrointestinal and renal risk, respiratory depression, addiction, nausea, constipation, cognitive impairment, mental problems, etc. [9,10]. Long-term large dosage always leads to opioid tolerance and opioid-induced pain sensitivity, then results in an opioid overdose crisis [9,10,11]. Limited by maximum dosage, opioid use has only accomplished around 30% pain relief on average [5,12]. Great efforts are still taken by researchers to discover new analgesics with low abuse potential.

Recent advances in the neurobiology of pain provide nonopioid targets for the development of innovative analgesics [13,14]. Targets such as voltage-gated sodium channels (Na_v_1.7 and Na_v_1.8), voltage-gated potassium channels (K_v_7), neurotensin receptor, angiotensin II receptor and α9α10 nicotinic acetylcholine receptor (nAChR) have been proved to be associated with pain and show analgesic effects with nonopioid mechanisms of action [14]. Screening compounds targeting these nonopioid targets has become a promising strategy of searching for novel analgesic candidates with low addiction potential.

Conotoxins are a class of active polypeptides that can specifically target a wide range of receptors, such as voltage-gated ion channels (sodium ion channel, calcium ion channel, potassium ion channel), nAChR, serotonin receptor, γ-aminobutyric acid receptor, G protein-coupled receptor (GPCR), neurotransmitter transporters, etc. [15,16]. Transcriptomic and proteomic studies of various species of marine cone snails indicated that each cone snail specimen can secrete about 1000–2000 different conopeptides [17]. Little overlap exists between different species of cone snails [18]. More than 100 million conopeptides were estimated in the conotoxin libraries. From this point of view, conotoxin libraries have been considered as a good source to find novel analgesic leads. However, more than 99% of the conotoxins have not been structurally and pharmacologically characterized.

In our work of screening analgesic peptides from the conotoxin libraries established by gene cloning from diverse *Conus* species, we decoded a peptide sequence of SS**C**GYLGQH**CC**IIPKHAY**C**YGYLE**C**NNRAV**C**V from a *Conus lividus* specimen. This sequence was identical to the mature region of the cDNA-deduced precursor (QGV13653.1) cloned by Mahardika’s group [19]. We named it Lv32.1 (LvXXXIIA) since it was the first conotoxin with framework XXXII (C-CC-C-C-C) reported from *Conus lividus*. Considering the novelty of its cysteine framework XXXII, in this work, we reported the synthesis, disulfide identification and analgesic characterization of conotoxin Lv32.1.

## 2. Results

### 2.1. Peptide Synthesis and Oxidation Optimization

The molecular weight of linear Lv32.1 was calculated to be 3569.34, which was consistent with the theoretical value, by ion of m/z 893.34 ([M+4H]^4+^) (Figure 1). The sequence of linear Lv32.1 was confirmed by a MALDI-TOF-MS/MS experiment (Figure 2). The fragments of GYL, HCCII and HAYCYGYLECNN were confirmed by the serial *y*_5_ − *y*_17_, *y*_19_ − *y_24_*, *b*_3_ − *b*_6_ and *b*_9_ − *b*_13_ ions. The observation of fragment ion SCG (*y_b_* m/z 248.0700) and ion *b*_3_ (m/z 278.0805) indicated the existence of fragment SSCG. Ions of NNRA (*y_b_* m/z 456.2314), RAVC (*y_b_* m/z 430.2231) and ion *y*_2_ (m/z 221.0954) suggested the sequence of NNRAVCV. Ion *y_b_* represented the *y* ion of the fragment, while *y_a_* was the ion whose C-terminal amino acid residue was decarboxylated. The ions of YLGQH (*y_a_* m/z 571.2987), LGQHCC (*y_b_* m/z 642.2487) and IIPKHA (*y_b_* m/z 660.4192) combined with the fragments of SSCG, GYL, HCCII, HAYCYGYLECNN and NNRAVCV undoubtedly identified the synthetic linear Lv32.1 to be SS**C**GYLGQH**CC**IIPKHAY**C**YGYLE**C**NN RAV**C**V.

The molecular weight of folded Lv32.1 was determined as 3563.20 by m/z 891.80 ([M+4H]^4+^ ion) (Figure 1). The mass difference between linear and folded Lv32.1 was 6 Da (Figure 1), which meant all of the six cysteines in Lv32.1 were oxidative to form three disulfide bridges. The linear Lv32.1 was folded by one-step oxidation since the typical disulfide connectivity for cysteine framework XXXII was unknown. In order to raise the production of folded Lv32.1 for subsequent study, the oxidation conditions on buffer, oxidizing agent and reaction time were optimized. The result showed that the folding process failed when the oxidizing agent was air, H_2_O_2_ (3%) and K_3_Fe(CN)_6_ (2 mg·mL^−1^), respectively. The linear Lv32.1 was folded only in the presence of GSH/GSSG. HPLC graphs of folding products of Lv32.1 in 0.1 M NH_4_HCO_3_ buffer for different reaction times were shown in Figure 3, revealing that a single predominant isomer was yielded. Different combinations of oxidative buffer with diverse concentrations of GSH/GSSG for progressive reaction times exhibited different impacts on the yields of the folded product (Figure 4 and Figure 5). The results showed that the productions of folded Lv32.1 in NH_4_HCO_3_ buffer were found to be generally higher than in other buffers, even though they had different concentrations of GSH/GSSG. The folding rate reached the maximum (34.94% ± 0.96%) in the condition of 0.1 M NH_4_HCO_3_ buffer with 1 mM GSH/0.5 mM GSSG at 25 °C for 1 h, in which the folded Lv32.1 was synthesized for subsequent study.

### 2.2. Novel Disulfide Connectivity of I–III, II–VI, IV–V

The six cysteines (Cys) in conotoxin Lv32.1 were freely connected to form three disulfide bridges, the connectivity of which remained unknown, since the one-step folding strategy. Herein, conotoxin Lv32.1 was stepwise reduced and subsequently alkylated to generate the two-cysteine-alkylated product (Figure 6, MW 3678.72) and four-cysteine-alkylated product (Figure 6, MW 3792.56). These two products were totally reduced and were subjected to MALDI-TOF-MS/MS sequencing (Figure 7 and Figure 8), respectively, to determine the disulfide connectivity of Lv32.1. For the MS/MS spectrum of the two-cysteine-alkylated product (Figure 7), the mass difference of *y*_23_/*y*_22_ (m/z 2749.2176/2589.1869) was 160.0307, which was equal to the mass of ethyl-labeled cysteine, since IAM (iodoacetamide) was applied as an alkylated agent. It meant the cysteine at position 10 was ethylated. The observation of ion *y*_5_ (m/z 604.3235) and fragment ion RAVC_(IAM)_ (*y_a_* m/z 459.2496) suggested that thiols in 31-Cys was also IAM-labeled. All the mass differences of *y*_30_/*y*_29_ (m/z 3507.5346/3404.5254), *y*_22_/*y*_21_ (m/z 2589.1869/2486.1778) or *b*_11_/*b*_10_ (m/z 1196.4282/1093.4190), *y*_14_/*y*_13_ (m/z 1663.7025/1560.6934) and *y*_8_/*y*_7_ (m/z 935.4186/832.4094) were 103.0092, 103.0091 (or 103.0092), 103.0091 and 103.0092, respectively, indicating that the thiols of 3-, 11-, 19- and 25-cysteines were in their reduced form. Herein, a disulfide linkage existed between Cys-II (10-Cys) and Cys-VI (31-Cys). As for the four-cysteine-alkylated product of Lv32.1 (Figure 8), the fragment ions of HC_(IAM)_C_(IAM)_I (*y_a_* m/z 656.3007) and QHC_(IAM)_C_(IAM)_I (*y_b_* m/z 699.2701) combined with ions *b*_11_ (m/z 1310.4711) and *a*_11_/*a_10_* (m/z 1282.4762/1122.4455) indicated that 10- and 11-Cys were ethylated. Cysteine at positions 3 and 31 turned out to be IAM-labeled due to the presence of ions *y*_2_ (m/z 278.1169) and *b*_4_ (m/z 392.1234) and fragment ion SC_(IAM)_G (*y_a_* m/z 277.0965). The detection of ions *y*_8_/*y*_7_ (m/z 935.4186/832.4049) or *b*_25_/*b*_24_ (m/z 2964.2394/2861.2303), and the fragment ion YC_(IAM)_Y (*y_a_* m/z 402.1482) revealed that 25-Cys and 19-Cys were the two cysteines unlabeled by IAM, revealing that 19-Cys (Cys-IV) and 25-Cys (Cys-V) were connected to form a disulfide bond, while the remaining four alkylated Cys (3-, 10-, 11- and 31-Cys) belonged to other groups. Considering Cys-II (10-Cys) and Cys-VI (31-Cys) had been confirmed to be linked by the two-cysteine-alkylated product (Figure 7), then the disulfide connectivity of Lv32.1 was characterized to be 1–3, 2–6, 4–5 (I–III, II–VI, IV–V), which was a rare linkage pattern and was first reported for the conotoxins with Cys framework XXXII.

### 2.3. Lv32.1 Exhibited Analgesic Potency

To further explore whether conotoxin Lv32.1 has an analgesic effect, we used a hot plate assay to detect the latent value of mice at different time points after the intracerebroventricular (*i.c.v.*) administration of Lv32.1 (10 nmol per mouse). The behavioral manifestation in the hot plate test showed an evident increase in the pain threshold with time. Figure 9 suggested that the analgesic effect of Lv32.1 began to appear at 60 min and it had a significant difference in latency compared with the saline group at 120 min (*p* < 0.01). This result stated that Lv32.1 exhibited a potent effect, which was a topic worthy of in-depth studying.

### 2.4. Weak Inhibition on Na_v_1.8 Channel and α9α10 nAChR

In order to investigate the analgesic target of conotoxin Lv32.1, we evaluated its inhibitory effect on the Na_v_1.8 channel and α9α10 nAChR. For the Na_v_1.8 channel, we tested the effect of Lv32.1 on the two states of the channel, namely the complete resting state (TP1) and the semi-activated state (TP2). In Figure 10, conotoxin Lv32.1 showed a weak effect on Na_v_1.8 channel by inhibiting (21.48 ± 1.22)% and (56.13 ± 0.05)% of the currents in the complete resting state (TP1) and the semi-activated state (TP2), respectively, at a concentration of 10 μmol/L. As to the effect on α9α10 nAChR, only about 10% of the ACh-evoked currents were inhibited by Lv32.1 with a final concentration of 1 μmol/L (Figure 11). Inhibitory effects on other subtypes of nAChR such as α3β4 nAChR were tested as well. However, no inhibition was detected (Figure 11).

## 3. Discussion

In our work of screening analgesic peptides from the conotoxin libraries established by gene cloning from diverse *Conus* species, we decoded a peptide sequence of SS**C**GYLGQH**CC**IIPKHAY**C**YGYLE**C**NNRAV**C**V from a *Conus lividus* specimen. The cDNA-deduced precursor (QGV13653.1) of this sequence has been reported by Mahardika’s group since 2019 [19]. To date, only ten mature peptides or precursors of conotoxins with cysteine framework XXXII have been found (Table 1), and their bioactivities were uncharacterized. Considering the novelty of cysteine framework XXXII (C-CC-C-C-C), we named the sequence Lv32.1 (or LvXXXIIA) and started the study on its synthesis, disulfide connectivity and analgesic effect.

The first step was to obtain a sufficient sample through peptide synthesis. Disulfide bonds play a significant role on maintaining and stabilizing the spatial conformation of peptides. Differences in disulfide connectivity, no matter the linkage number or pattern, may result in diverse effects on targets. Conotoxin Lv32.1 contains six cysteines that can form three disulfide bridges. Theoretically, Lv32.1 have 15 isomers with different disulfide connectivities. How to obtain one single three-disulfide isomer of Lv32.1 efficiently was of great importance. In our experience [26], for disulfide-rich conotoxins, one-step (direct) folding generally turns out to be more efficient than regioselective oxidation since it has fewer reaction steps and purification operations. Although a variety of disulfide isomers may generate, conotoxins tend to fold into one or two predominant bioactive isomers with small amounts of by-products. Luckily, a single predominant isomer was yielded with direct folding only in the presence of GSH/GSSG, while it failed when oxidized by air, H_2_O_2_ and K_3_Fe(CN)_6_, respectively, indicating that GSH/GSSG played a significant role in the direct folding of Lv32.1. Then, we investigated the folding condition and found that the production in NH_4_HCO_3_ buffer was larger than in other buffers (Tris-HCl, NH_4_HCO_3_, NH_4_OAc, NaCl-EDTA·2Na, PBS, Na_2_HPO_4_/NaH_2_PO_4_). As for the concentration of GSH/GSSH, it exhibited different impacts on different buffers. The 1 mM GSH/0.5 mM GSSG yielded the most in 0.1 M NH_4_HCO_3_ buffer. With regard to the reaction time, we conducted the dynamic detection of the folding process with HPLC−MS analysis (Figure 3 and Figure 5). However, the production declined after 1 h in 0.1 M NH_4_HCO_3_ buffer, while the decline generally happened after 3 h in other buffers (Figure 5). Herein, investigation on the dynamic variation during the folding process was quite necessary and helpful. We speculated that the reason for the production decline may be the aggregation of the folding peptide over time. In NH_4_OAc buffer, the production of reacting for 3 h to 12 h was relatively stable. Maybe NH_4_OAc buffer could weaken the aggregation effect. The results (Figure 3, Figure 4 and Figure 5) showed that the maximum yield reached up to (34.94 ± 0.96)% when Lv32.1 was folded in 0.1 M NH_4_HCO_3_ buffer with 1 mM GSH/0.5 mM GSSG at 25 °C for 1 h. The optimized condition provided an efficient solution for the synthesis of Lv32.1.

A previously found M-superfamily conotoxin Mo3964, whose disulfide connectivity was determined to be 1–3, 2–5, 4–6 [22], was the first conotoxin with Cys framework XXXII that has been structurally characterized. In this work, the disulfide linkage of the synthetic Lv32.1 was identified to be a rare pattern of 1–3, 2–6, 4–5, which was first reported for the conotoxins with Cys framework XXXII and different from the common connectivities established for conotoxins with six cysteines (Table 2) [16].

The hot plate assay is one of the most commonly used tests for evaluating the analgesic efficacy in rodents [37]. The pain endurance is expressed by the latency of the mice to raise and lick hind paws, to flutter or to jump up. The behavioral manifestation in the hot plate test showed an evident increase in the pain threshold with time-dependency, which illustrated that Lv32.1 preliminarily exhibited analgesic potency. More evidence and more studies were still needed. Nonopioid receptors such as the Na_v_1.8 channel and α9α10 nAChR have become high-profile targets to develop novel analgesics [14,38,39]. Na_v_1.8 is generally distributed in DRG (dorsal root ganglion) neurons, and it is considered as an important analgesic target that is of great influence on the transmission of action potentials in nociceptors [39]. Conotoxin μ-MrVIB, a selective Na_v_1.8 blocker, alleviated allodynia and hyperalgesia in both neuropathic and chronic inflammatory pain models [40]. The α9α10 nAChR has also been proposed as a target for the treatment of neuropathic pain. The α9α10 nAChR antagonist conotoxin GeXIVA reduced mechanical hyperalgesia in the rat chronic constriction injury model of neuropathic pain [38] and also alleviates and reverses chemotherapy-induced neuropathic pain [41]. Herein, we detected the inhibitory effect of Lv32.1 on the Na_v_1.8 channel and α9α10 nAChR, but weak inhibition was observed. Activities on other analgesic targets, such as the Na_v_1.7 channel, voltage-gated calcium channels (Ca_v_) and neurotensin receptor, remains to be studied.

## 4. Materials and Methods

### 4.1. Peptide Synthesis and Oxidation Optimization

Conotoxin Lv32.1 was obtained by linear peptide synthesis and a one-step oxidation strategy [26]. Briefly, the linear peptide was synthesized on the Wang Resin by standard Fmoc (*N*-9-flurenylmethoxycarbonyl) solid-phase method. The Fmoc-amino acids were coupled with HBTU/DIEA and then the Fmoc groups were removed by Piperidine/DMF (v:v 20:80). The six cysteine residues were protected by triphenylmethyl (trt) groups and then deprotected at the peptide release process. Linear peptide was released from the resin by cleavage mixture (TFA/thioanisole/phenol/EDT/H_2_O, v:v:v:v 87.5:2.5:2.5:2.5) at room temperature for about 2 h. Crude peptide was precipitated and washed by cold ether (0 °C) and purified by pre-HPLC to gain pure linear peptide, which was subjected to MALDI-TOF-MS/MS analysis (protocol was the same as that in the Section 4.2) to verify the sequence.

The one-step oxidation conditions for the oxidizing agent {air, GSH/GSSG, 3% H_2_O_2_, 2 mg·mL^−1^ K_3_Fe(CN)_6_}, buffer (0.1 M Tris-HCl, 0.1 M NH_4_HCO_3_, 0.1 M NH_4_OAc, 75 mM NaCl-EDTA·2Na, PBS, 30 mM Na_2_HPO_4_/NaH_2_PO_4_), concentration of GSH/GSSG (0.5 mM/0 mM, 0.5 mM/0.5 mM, 1 mM/0.5 mM, 2.5 mM/0.5 mM) and reaction time (5 min, 15 min, 0.5 h, 1 h, 2 h, 3 h, 4 h, 6 h, 9 h, 12 h) were optimized. The other conditions, such as the concentration of linear Lv32.1 (50 μM), folding temperature (25 °C) and 1 mM EDTA added to the buffer, were applied. The folded product with three disulfide bridges was monitored by UPLC-TQD-MS. Yields of folded Lv32.1 in different conditions were calculated as $\frac{Peak\;area_{folded\;peptide}}{Peak\;area_{linear\;peptide}}\times 100\%$. Considering all the conditions, the linear Lv32.1 (50 μM) was folded in a 0.1 M NH_4_HCO_3_/1 mM EDTA buffer with GSH/GSSG (1 mM/0.5 mM) at 25 °C for 60 min. The folded Lv32.1 was purified by a HPLC system for subsequent study.

### 4.2. Identification of Disulfide Connectivity

Folded Lv32.1 was stepwise reduced and subsequently alkylated as previously reported [26,42]. The reducing reagent was prepared by dissolving 1.7 mg TCEP in citrate buffer (0.1 M, pH 3.0). The folded peptide (500 μg in 200 μL ddH_2_O containing 0.1% TFA) was mixed with 100 μL reducing reagent at 40 °C for 2 h. The reaction mixture was constantly monitored by UPLC-TQD-MS to confirm the generation of a one-disulfide-reduced product and two-disulfide-reduced product. Then, the reduced mixture was blended with a 100 μL alkylation solution (9.24 mg iodoacetamide dissolved in 25 μL acetonitrile and 75 μL 1 mM pH 8.0 Tris-HCl buffer) at 25 °C for 12 h in the dark. The two-cysteine-alkylated product and the four-cysteine-alkylated product in the alkylating mixture were verified by UPLC-TQD-MS detection. Then, the two alkylated products were individually reduced by TCEP to obtain nondisulfide products, namely products I and II, which were determined by MALDI-TOF-MS/MS (Ultraflextreme, Bruker, Massachusetts, Germany) to verify which pair of thiol groups was synchronously alkylated. Products I and II were severally loaded by successively dropping 1 µL of sample solution (20 μg peptide in 50% acetonitrile aqueous solution) and 1 µL of matrix solution (1 mg HCCA in 47.5 µL ddH_2_O, 50 µL acetonitrile and 2.5 µL TFA) onto the same circle on the sample plate and then dried off. The parameter settings for MALDI–TOF–MS/MS analysis by FlexControl (Bruker, Massachusetts, Germany) were shown as follows: laser frequency 1000 Hz, laser energy range 20–40%, ion source voltage 7.5 kV, Lift 1 voltage 19.00 kV and Lift 2 voltage 2.80 kV. The ion masses and related fragmentation masses were marked by FlexAnalysis software (Bruker, Massachusetts, Germany) and were searched and matched by Mascot search at http://www.matrixscience.com/cgi/nph-mascot.exe?1 (accessed from 10 November 2021 to 30 October 2022). Mascot searching was performed on the NCBI and Swiss-prot databases. The mass tolerance and fragment mass tolerance were set to be ±0.1 Da and ±0.5 Da, respectively. Significance threshold *p* value was set as <0.05.

### 4.3. Analgesic Evaluation by Hot Plate Test in Mice

#### 4.3.1. Animals

Kunming mice (male, 18–22 g) were purchased from Hunan SJA Laboratory Animal Co., Ltd. (Changsha, China) with the permit SCXK 2019-0004. Six mice were housed per cage and kept at a 23 ± 1 °C humidity-controlled environment and associated with a 12-h light/dark cycle light at 8 a.m. Experiments were performed during the light cycle. All the experiments were conducted in accordance with IASP guidelines on the use of awake animals with efforts made to minimize the number of animals and their discomfort [43].

#### 4.3.2. Intracerebroventricular (*i.c.v.*) Surgery

Mice were anesthetized with sodium pentobarbital (75 mg/kg, *i.p.*) and hair was shaved off the head to expose the skin. Then mice were placed on a stereotaxic frame adaptor comprising adjustable ear bars and a tooth holder. The skin over the skull was opened and 10% hydrogen peroxide was applied to peel off the periosteum. Stereotaxic coordinates were measured by a stereoscopic brain locator (Rayward Life Technology Co., Ltd., Shenzhen, China) from Bregma and were based on The Mouse Brain in Stereotaxic Coordinates. Skull holes were pierced 1.3 mm to the right of the sagittal suture, 0.6 mm back to the fontanel and the depth was 2.0 mm. Then, the catheter was gently placed into the drilled hole, and mixed dental cement powder was used to fix it. After skin suture, penicillin dry powder was applied to the wound to prevent infection, and the catheter cap was put on. Animals were kept one per cage after they recovered from anesthesia with free access to water and food. Mice were screened three days after surgery for subsequent experiments. On the morning of the test, a 26-gauge needle and a polyurethane cannula were inserted into the lateral ventricle. The needle was left for 2 min after injection to ensure drug delivery.

#### 4.3.3. Pain Threshold Measurement by Hot Plate Test

The pain threshold was measured using a hot plate detector (IITC33, Woodland Hills, CA, USA), which consisted of a rectangular heated metal plate surrounded by Plexiglas. To measure the somatic pain response, mice were placed in a cylindrical glass bucket approximately 10 cm wide on the hot plate that was heated to 55 ± 0.5 °C. The reaction time was recorded when animals began to exhibit signs of pain avoidance such as jumping or paw licking. Conotoxin Lv32.1 was dissolved in saline (0.9% sodium chloride) to obtain the stock solution (final concention of 2 mM), which was administered by *i.c.v.* injection in a volume of 5 μL (10 nmol) per mouse (n = 9). The control group (n = 5) was set by injection of saline (0.9% sodium chloride, 5 μL per mouse).

#### 4.3.4. Statistical Analysis

Data were expressed as mean ± S.E.M. and analyzed using Graphpad Prism 6.0 (GraphPad Software, San Diego, CA, USA). For the hot plate experiment, data were analyzed by two-way ANOVA and post hoc Bonferroni’s multiple comparison analyses.

### 4.4. Whole-Cell Patch Clamp Measurement on Na_v_1.8 Channel

The blocking effect of Lv32.1 on the Na_v_1.8 channel was performed, as reported [26], by whole-cell patch clamp technique [44]. In short, the Na_v_1.8 expressed CHO cells (6.5 × 10^3^), established by Beijing Asip Biotechnology Co., Ltd. (Beijing, China), were planted on coverslips in 24-well plates and cultured for 18 h before patch clamp recording. Extracellular fluid solution was prepared with 140 mM NaCl, 3.5 mM KCl, 1 mM MgCl_2_•6H_2_O, 2 mM CaCl_2_•2H_2_O, 10 mM *D*-Glucose, 10 mM HEPES and 1.25 mM NaH_2_PO_4_•2H_2_O, adjusted to pH 7.4 with NaOH. Conotoxin Lv32.1 solution (10 mM, dissolved with DMSO) was diluted to be 10 µM by extracellular fluid solution. A selective Na_v_1.8 channel blocker A-803467 [5-(4-chlorophenyl-*N*-(3,5-dimethoxyphenyl)furan-2-carboxamide, 10 µM in extracellular fluid solution] was purchased from Beijing Enokai Technology Co., Ltd. (Beijing, China) and used as the positive control. For current recording, the cell-planted coverslip was placed into the recording bath of an inverted microscope (MP285, Sutter Instrument, Novato, CA, USA) and the recording electrode was contacted to the cell. After sealing the cell, the membrane voltage was clamped at −120 mV for 30 ms (microseconds) as the resting state (TP1 state). The holding potential depolarized to 0 mV for 50 ms and was recovered to −75 mV (semi-inactivation voltage) for 8 s to reach the half-inactivated state (TP2 state). Solutions of Lv32.1, positive control and negative control (DMSO) separately passed through the recording bath by gravity perfusion for 5 min, repeated thrice, at both TP1 state and TP2 state. The corresponding currents were recorded by an EPC-10 amplifier (HEKA, Reutlingen, Baden Wurdenburg, Germany) and stored by PatchMaster (HEKA, Reutlingen, Baden Wurdenburg, Germany) software.

### 4.5. Two-Electrode Voltage Patch Clamp Recording on α9α10 nAChR

The α9α10 nAchR inhibitory activity of Lv32.1 was conducted as previously reported by two-electrode voltage patch clamp system (Molecular Devices, CA, USA) [45]. Solutions were prepared as follows: ND96 buffer (96.0 mmol/L NaCl, 1.8 mmol/L CaCl_2_, 2.0 mmol/L KCl, 1.0 mmol/L MgCl_2_, 5.0 mmol/L HEPES, pH 7.4), ND96 buffer containing antibiotic (10 mg/L penicillin, 10 mg/L streptomycin and 100 mg/L gentamicin) and OR-2 buffer (82.5 mM NaCl, 2.0 mM KCl, 1.0 mM MgCl_2_·6H_2_O, 5 mM HEPES, pH 7.5). Briefly, oocytes were obtained by dissecting from pre-anesthetized mature female *Xenopus laevis* frogs. After enzymolysis by 20 mg collagenase (1.5 mg/mL, (Sigma Aldrich, St. Louis, MO, USA) incubated in 40 mL OR-2 buffer at 25 °C for 40 min, separated oocytes were gained for the subsequent expression of α9α10 nAchR. Linearized plasmids of the rat α9 and α10 subunits (Genebank numbers: NM_001081104, NM_001081424) were transcribed in vitro using the Mmessage MESSAGE mMACHINE^TM^ SP6 Transcription Kit (Thermo Fisher Scientific CN, Shanghai, China). Then, α9 and α10 cRNAs were purified by a MEGAclear^TM^ Transcription Clean-Up Kit (Thermo Fisher Scientific CN, Shanghai, China). The cRNAs of α9 and α10 were mixed at a ratio of 1:1 and injected into oocytes (59.6 nL cRNA for each subunit) and cultured at 17 °C for 2–3 days. The oocytes were then fixed in an oocyte chamber and perfused with BSA-containing ND96 at a rate of 2 mL/min. The oocyte was given 100 μM ACh for 2 s and ND96 for 58 s. The membrane potential of the oocyte was clamped at −70 mV by two-electrode voltage patch clamp and the elicited current was recorded as control current. Conotoxin GeXIVA [1,2] was used as the positive control in our previous study [46]. When the current stayed stabilized, 5 μL of 10 μmol/L Lv32.1 solution (dissolved in ND96 buffer) was added into the 50 µL chamber and incubated for 5 min to record the current. The measurement was repeated on 6 oocytes.

### 4.6. Data Analysis of Electrophysiological Assay on α9α10 nAChR and Na_v_1.8 Channel

Data for α9α10 nAChR were collected by Clampfit 10.2 (Axon Instruments, Boston, MA, USA), and data for Na_v_1.8 Channel were analyzed by PatchMaster (HEKA, Reutlingen, Baden Wurdenburg, Germany) and IGOR Pro (Wave Metrics, Portland, OR, USA). The reaction ratio was defined as $\frac{Peak\;current_{conotoxin}}{Peak\;current_{control}}$, and the percentage of inhibition was determined as (1 − $\frac{Peak\;current_{conotoxin}}{Peak\;current_{control}}$ × 100%). The mean and standard deviation were calculated based on parallel experiments. GraphPad Prism 6.0 (GraphPad Software, San Diego, CA, USA) was used for data processing and graphing.

## 5. Conclusions

In this work, we highlight the efficient synthesis, novel disulfide linkage and analgesic potential of Lv32.1, which laid a positive foundation for further development of conotoxin Lv32.1 as an analgesic candidate.

## Figures and Tables

**Figure 1 molecules-27-08617-f001:**
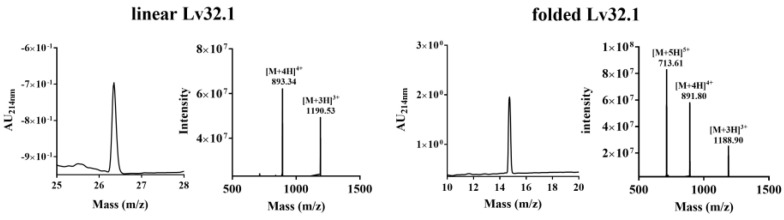
HPLC and LC-MS spectra of linear and folded Lv32.1.

**Figure 2 molecules-27-08617-f002:**
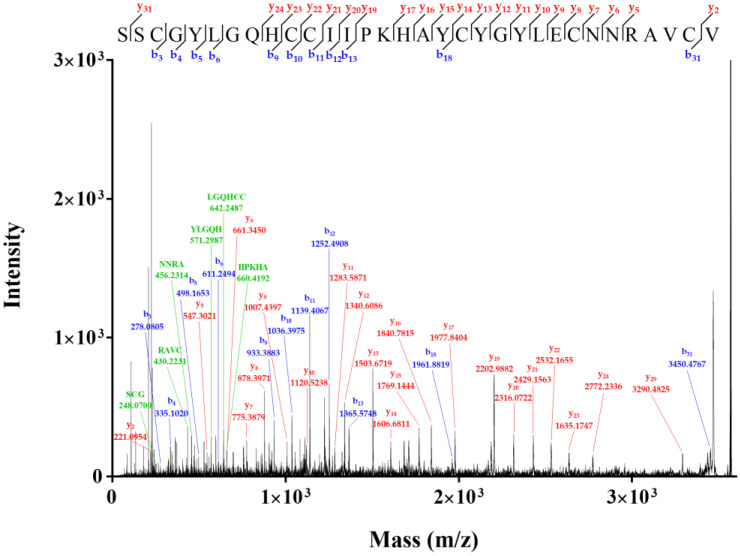
MALDI-TOF-MS/MS spectrum for sequencing of synthetic Lv32.1.

**Figure 3 molecules-27-08617-f003:**
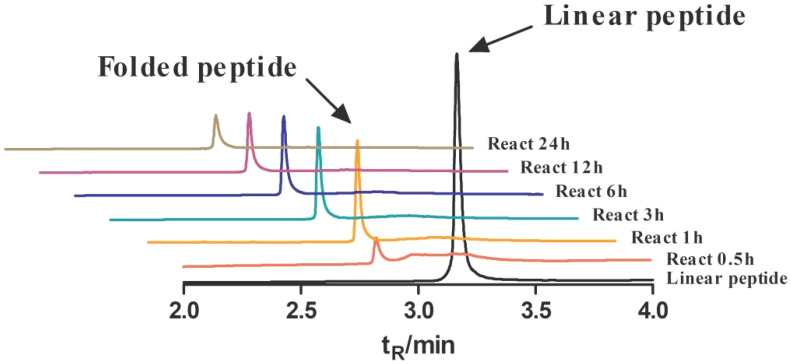
HPLC graphs of folding products of Lv32.1 in 0.1 M NH_4_HCO_3_ buffer with 1 mM GSH/0.5 mM GSSG for 0.5 h, 1 h, 3 h, 6 h, 12 h and 24 h.

**Figure 4 molecules-27-08617-f004:**
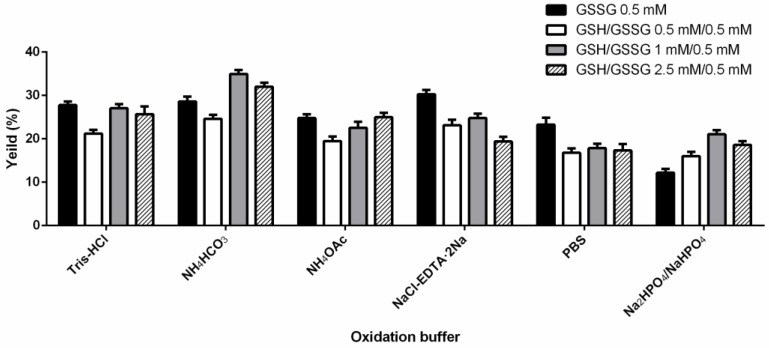
Yields of folded Lv32.1 in different buffers with diverse concentrations of GSH/GSSG for 1 h.

**Figure 5 molecules-27-08617-f005:**
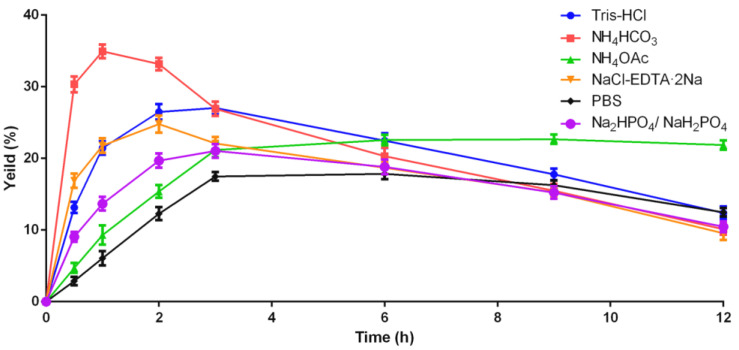
Yields of folded Lv32.1 in different buffers with GSH/GSSG (1 mM/0.5 mM) at progressive reaction times.

**Figure 6 molecules-27-08617-f006:**
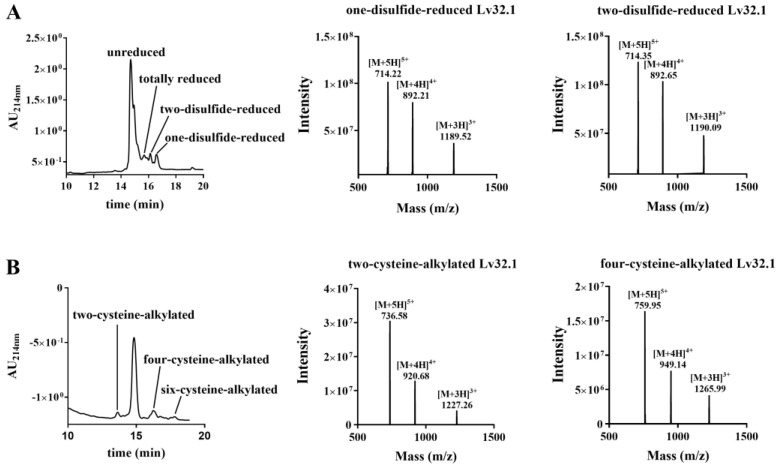
HPLC and LC-MS spectra of stepwise reduction by TCEP and follow-up alkylation by IAM. (**A**) HPLC and LC-MS spectra of stepwise reduction products of Lv32.1. (**B**) HPLC and LC-MS spectra of alkylated products of stepwise-reduced Lv32.1.

**Figure 7 molecules-27-08617-f007:**
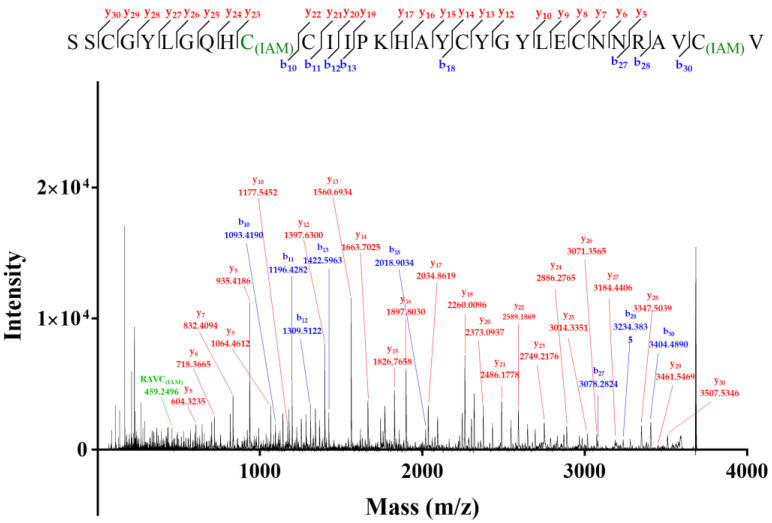
MALDI-TOF-MS/MS spectrum of two-cysteine-alkylated product of Lv32.1. C_(IAM)_ represented the cysteine alkylated by IAM.

**Figure 8 molecules-27-08617-f008:**
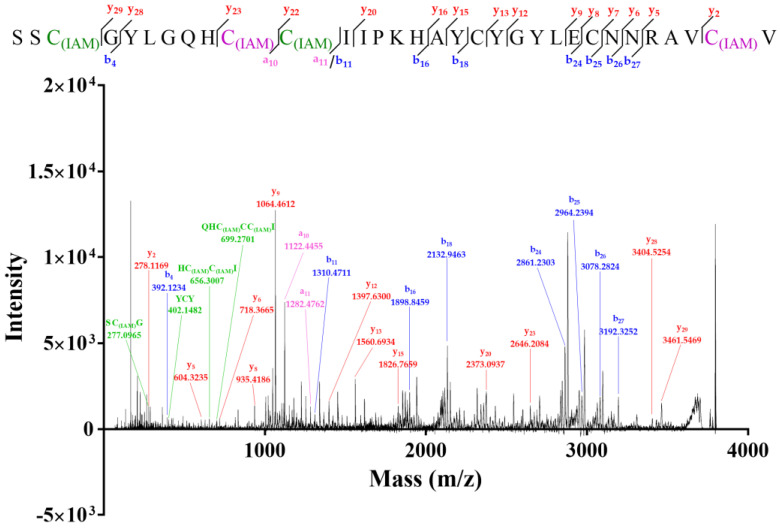
MALDI-TOF-MS/MS spectrum of four-cysteine-alkylated product of Lv32.1. C_(IAM)_ represented the ethyl-labeled cysteine by IAM.

**Figure 9 molecules-27-08617-f009:**
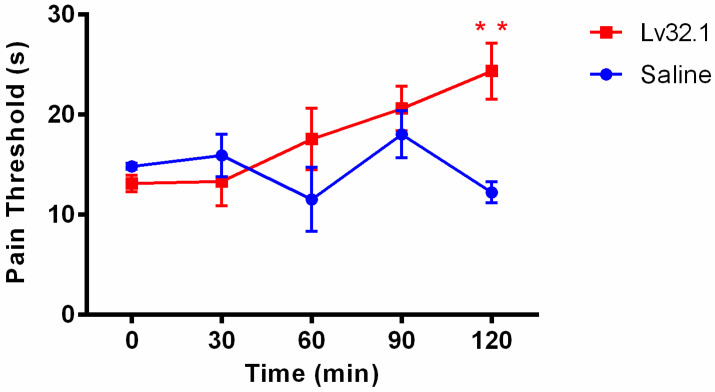
Analgesic effect of conotoxin Lv32.1 on hot plate assay. Each latency was shown as mean ± SEM. ** Meant that conotoxin Lv32.1 group (n = 9) had significant difference compared with the saline-vehicle control group (n = 5) (*p* < 0.01).

**Figure 10 molecules-27-08617-f010:**
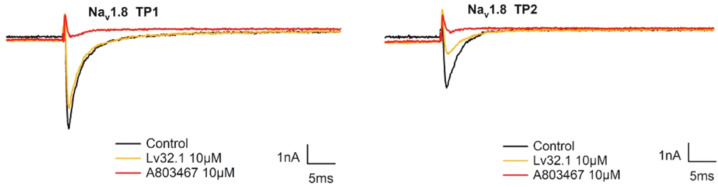
Traces of current inhibition on Na_v_1.8 channel by Lv32.1.

**Figure 11 molecules-27-08617-f011:**
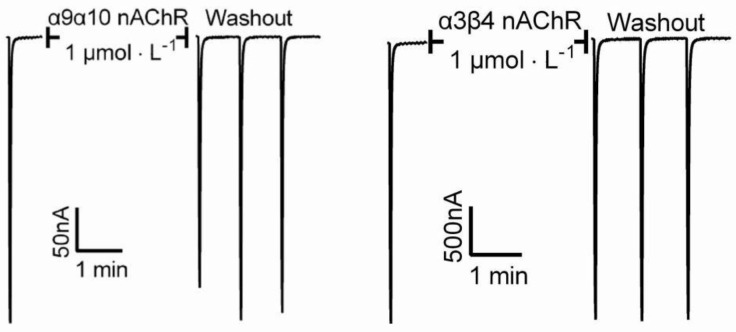
Inhibition of ACh-evoked currents mediated by α9α10 and α3β4 nAChRs.

**Table 1 molecules-27-08617-t001:** Reported conotoxins or precursors with cysteine framework XXXII.

Name	Source	Sequence	Reference
Lv32.1/ QGV13653.1	*C. lividus*	SS**C**GYLGQH**CC**IIPKHAY**C**YGYLE**C**NNRAV**C**V	This work [19]
Ba32.1	*C. bayani*	D**C**GERDEP**CC**VNSSGVKY**C**ESPWS**C**MHTTLL**C**EQN	[20]
Lt32.1	*C. litteratus*	SS**C**GYLGQQ**CC**IVPKRAY**C**HGDLE**C**NPVAM**C**VA	[21]
Mo3964	*C. monile*	DGE**C**GDKDEP**CC**GRPDGAKV**C**NDPWV**C**ILTSSR**C**ENP	[22]
Qc32.1	*C. quercinus*	SS**C**GYLGQP**CC**VVPRRAY**C**HGDLE**C**NDVTM**C**V	[23]
AMP44674.1	*C. betulinus*	SS**C**GYVGQP**CC**IVPRRAY**C**HGDLN**C**NNVAM**C**V	[24]
ACV87167.1	*C. eburneus*	SG**C**GYLGEP**CC**ISPKRAY**C**HGDLE**C**NNVAM**C**V	[25]
ACV87166.1	*C. marmoreus*	SG**C**GYLGEP**CC**VAPKRAY**C**HGDLE**C**NNVAM**C**V	[25]
ACV87168.1	*C. marmoreus*	SG**C**GYLGEP**CC**VAPKRAY**C**HGDLE**C**NSVAM**C**V	[25]
ACV87169.1	*C. litteratus*	SG**C**GYLGEP**CC**VAPKRAY**C**HGDLE**C**NNIAM**C**V	[25]

**Table 2 molecules-27-08617-t002:** Common connectivities of cysteine frameworks with six cysteines.

Disulfide Connectivity	Cysteine Framework	Reference
1–2, 3–4, 5–6	XXIII	[27]
1–3, 2–5, 4–6	IV, XXVII, XXXII	[22,28,29]
1–4, 2–5, 3–6	III, VI/VII	[30,31,32]
1–6, 2–4, 3–5	III	[33]
1–5, 2–3, 4–6	IV	[34]
1–5, 2–4, 3–6	III	[35,36]
1–3, 2–6, 4–5	XXXII	this work

## Data Availability

Data are contained within the article.

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
