# Peer review of "Synthesis and Characterization of an Analgesic Potential Conotoxin Lv32.1"

_molecules, 2022, doi:10.3390/molecules27238617_

Round 1
Reviewer 1 Report
This manuscript reports the chemical synthesis and analgesic activity of Conotoxin Lv32.1 derived from the Conus lividus. The optimization of oxygen folding condition of lv32.1 was conducted in this work. The analgesic function analysis was carried out using the hotplate test of Kunming mice. And this work try to describe the receptor of lv32.1 by measuring the modification on Nav1.8 and the subtype of nAChR α9α10 using Patch Clamp, however, the results was not obvious. This work should be improved before considering acceptable.
1. In this work, the title of part two is “2. Results and Discussions” and part three is “3. Discussions”, “Discussions” is repeated.
2. Most of the contents and expressions of “3. Discussions” are repeated with the methods and results of experiments.
3. The result of Figures 8, it will be better to add a positive control. For example, one of drugs used in clinic or a kind of conotoxin which was known as analgesic function.
It will be better to indicate The Y-axis as “pain threshold” or “percentage of pain threshold increasing”.
4. Figure 9 and Figure 10, the results of positive control should be added.
5. Table 2, the reference should be noted in the end of each line.
Author Response
Dear Reviewer:
Thanks for your valuable advises.
Our answers to your comments are as follows:
(1)The error was revised.
(2)The repetitive descriptions were deleted and more discussions were supplemented.
(3)Morphineis a commonly used positive control for analgesic experiment. But it’s difficult to obtain even for scientific experiments since complex special reviews and reports were needed. Herein, we did not use a positive control and primarily aimed to investigate whether Lv32.1 would increase the pain threshold when comparing with the control.
The Y-axis of Figure 8 was changed to “pain threshold”.
(4)The trace of the positive control A803467 for Nav1.8 was added to Figure 10. The trace of the positive control, a selective a9a10 nAChR blocker conotoxin GeXIVA[1,2], for a9a10 nAChR was shown in our previous work (Luo, sulan et al. Structure and Activity Studies of Disulfide-Deficient Analogues of alphaO-Conotoxin GeXIVA. DOI: 10.1021/acs.jmedchem.9b01409). The information was supplemented to the method.
(5)The references were attached to Table 2 as required.
Thanks again and wish you a good day.
Sincerely,
Dr. Ying Fu
Reviewer 2 Report
In this paper, authors have synthesized a new conotoxin Lv32.1 with a new cysteine framework XXXII, and determined its disulfide bond connectivity and analgesic activity, but its targets is still unknown though they found that Lv32.1 displays a low activity against Nav1.8 channel, α9α10 and α3β4 nAChRs. However, I think this work is still interesting because its disulfide bond connectivity of the cysteine framework XXXII in Lv32.1 was characterized. Some revisions or explanations should be made:
1. The authors should provide some folding figures of Lv32.1 in the optimized conditions.
2. Generally, the linear conotoxins with six cysteine residues fold to aim products for 16-48 h in the presence of GSH/GSSG, but Fig.4 showed that aim products formed in 2-4 h and were unstable (the yields of folded products decreased), could you give some explanations?
3. Generally, the number of mouse in animal experiments is 8-10, but only five rats were used in Figure 8.
4. Line 154-155, “the intracerebroventricular (i.c.v.) administration of Lv32.1 (10 nM)” is not clear, 10 nmol/kg? Please clarify it. High doses of Lv32.1 should be included in the experiments.
Author Response
Dear Reviewer:
Thanks for your valuable advises.
Our answers to your comments are as follows:
(1)Folding figures of Lv32.1 in the optimized conditionswere shown in Figure 3.
(2)In our experience, for conotoxins with three disulfide bridges, the desired folded peptide could be generated in 2 hours in the presence of GSH/GSSH by one-step oxidation. We also detected the reaction for over 24h, but the prodution was far less than the prodution for 2h (shown in Figure 3). Herein, dynamic detection for the folding process was quite necessary and helpful. We speculated that the reason for the production decrease may be the aggragation of the folding pepetide over time. These information was added to the section “discussions”.
(3)Due to the limitation of experimental conditions, only 5 mice were available for the controlgroup at that time. Luckily, this 5 mice provided adequate useful data for the control group to show a significant difference between the two groups.
(3)The intracerebroventricular (i.c.v.) administration of Lv32.1 was 10 nmol per mouse, not 10 nmol/kg, which was clarified in the method and in the result.
Thanks again and wish you a good day.
Sincerely,
Ying Fu
Round 2
Reviewer 1 Report
This version has been modified according to the comments.